

# Above- and below-ground trait coordination in tree seedlings depend on the most limiting resource: a test comparing a wet and a dry tropical forest in Mexico

Lucía Sanaphre-Villanueva[1,2], Fernando Pineda-García[3], Wesley Dáttilo[4], Luisa Fernanda Pinzón-Pérez[2], Arlett Ricaño-Rocha[2] and Horacio Paz[2,5,6]

[1] Centro del Cambio Global y la Sustentabilidad A.C., Consejo Nacional de Ciencia y Tecnología, Villahermosa, Tabasco, México

[2] Instituto de Investigaciones en Ecosistemas y Sustentabilidad, Universidad Nacional Autónoma de México, Morelia, Michoacán, México

[3] Escuela Nacional de Estudios Superiores, Unidad Morelia, Universidad Nacional Autónoma de México, Morelia, Michoacán, México

[4] Red de Ecoetología, Instituto de Ecología, A.C., Xalapa, Veracruz, México

[5] Laboratorio Nacional de Innovación Ecotecnológica para la Sustentabilidad, Universidad Nacional Autónoma de México, Morelia, Michoacán, México

[6] Center for Stable Isotope Biogeochemistry and the Department of Integrative Biology, University of California, Berkeley, CA, United States of America

Corresponding author
Horacio Paz, hpaz@iies.unam.mx

## ABSTRACT

The study of above- and below-ground organ plant coordination is crucial for understanding the biophysical constraints and trade-offs involved in species' performance under different environmental conditions. Environmental stress is expected to increase constraints on species trait combinations, resulting in stronger coordination among the organs involved in the acquisition and processing of the most limiting resource. To test this hypothesis, we compared the coordination of trait combinations in 94 tree seedling species from two tropical forest systems in Mexico: dry and moist. In general, we expected that the water limitation experienced by dry forest species would result in stronger leaf-stem-root coordination than light limitation experienced by moist forest species. Using multiple correlations analyses and tools derived from network theory, we found similar functional trait coordination between forests. However, the most important traits differed between the forest types. While in the dry forest the most central traits were all related to water storage (leaf and stem water content and root thickness), in the moist forest they were related to the capacity to store water in leaves (leaf water content), root efficiency to capture resources (specific root length), and stem toughness (wood density). Our findings indicate that there is a shift in the relative importance of mechanisms to face the most limiting resource in contrasting tropical forests.

## INTRODUCTION

The different features of an organism do not represent stochastically independent dimensions, but rather they are correlated with one another through the interplay of genetic constraints and selective pressures (*Pavlicev, Cheverud & Wagner, 2009*). Moreover, the evolutionary maximization of one function is frequently attained by minimizing another, which is described as a functional trade-off. When plant traits are consistently correlated among species, they form axes or dimensions of trait variation (*Wright et al., 2006*), in which traits that are functionally or developmentally related to each other evolve in a coordinated fashion in response to selective pressures (*Pavlicev, Cheverud & Wagner, 2009*; *Machado et al., 2019*). For example, at a global scale, plants are differentiated by their traits along a trade-off between rapid acquisition vs conservation of resources (*Reich, 2014*).

Variation in functional traits reflects the adaptation of organisms to their abiotic environment. This idea has been tested by estimating the relationship between traits and some fitness components, such as survival or reproduction, which has shown a much stronger selective effect of the physical environment than biotic factors (*Caruso, Maherali & Martin, 2020*). Consequently, the relationships between specific plant functional traits with environmental conditions have demonstrated that certain characteristics have potential adaptive value. These include, for example, increased photosynthetic capacity and leaf nitrogen content (*Vogan & Maherali, 2014*), rooting depth (*Schenk & Jackson, 2002*), deciduousness (*Palomo-Kumul et al., 2021*) and cavitation resistance in dry environments (*Maherali, Pockman & Jackson, 2004*), and specific leaf area, leaf dry matter content (*Poorter, 2009*), leaf area, and plant height (*Guerin et al., 2022*) in mesic environments.

Although it is clear that plant strategies exist and respond to environmental selective pressures, it is currently debated whether this plant response occurs at an organismic scale, with their different organs tightly converging functionally (*Grime, 2006*; *Reich, 2014*), or whether each organ responds individually, resulting in independent axes of functional variation (*Baraloto et al., 2010*). Some studies have found coordination (*i.e.*, trait correlation or covariation) among root, stem, and leaf traits, supporting the idea that species have diversified across ecological strategies in response to environmental gradients resulting in a whole plant strategy (*de la Riva et al., 2016*; *Freschet et al., 2010*; *Ávila Lovera et al., 2022*). However, others have found that different tissues have diversified under independent selective pressures, or respond to independent phylogenetic constraints, resulting in decoupled organs (*Fortunel, Fine & Baraloto, 2012*; *Valverde-Barrantes, Smemo & Blackwood, 2015*; *Bowsher et al., 2016*; *Wang et al., 2017*). Thus, there is still no consensus. One possibility is that such a discrepancy may result from limited empirical evidence, and the fact that leaf and stem traits have been more extensively studied than root traits, leaving the relationships between root traits and their functions poorly understood (*Freschet et al., 2021*).

Tropical forests have been used as models to test plant coordination hypotheses because they have high species and functional diversity (*Vleminckx et al., 2021*). While strong coordination among leaf, stem and root traits has been found in dry forests (*Markesteijn et al., 2011a*; *Méndez-Alonzo et al., 2012*), but see (*Silva et al., 2018*)), studies in moist forests

have shown leaves and roots to be independent axes of variation (*Fortunel, Fine & Baraloto, 2012*; *Vleminckx et al., 2021*). Together, these findings suggest the theoretical expectation proposed by *Dwyer & Laughlin (2017)* that under less restricted conditions, many possible conformations of leaf, stem, or roots are biophysically possible and ecologically successful (*Dwyer & Laughlin, 2017*; *Santiago et al., 2018*). In addition, if coordination among traits results from natural selection purging uncoordinated variants, correlations would be expected mainly among the traits with a higher selective value in a given environment (not among any possible trait combination). Therefore, assessing the coordination of traits related to potentially relevant functions under contrasting environmental conditions could improve our understanding of functional coordination and its role in maintaining biodiversity.

In this study, we assessed whether coordination between above ground (leaf and stem) and below ground (roots) organs differ between seedlings of species from a dry versus a moist tropical forest in Mexico. The two forests differ in their most limiting resource: water in the dry forest, and light in the moist forest. We hypothesized that environmental restriction of a particular resource could result in stronger coordination among traits involved in the acquisition and processing of that resource (*i.e.*, the traits under the strongest selective pressure). Specifically, we expected that in the dry forest, where drought pulses impose a strong selective pressure for fine-tuned synchronization among above ground and below ground organs to acquire, transport, and use water, there would be stronger above–below ground trait coordination (*Markesteijn et al., 2011a*; *Méndez-Alonzo et al., 2013*; *Méndez-Alonzo et al., 2012*; *Paz, Pineda-García & Pinzón-Pérez, 2015*; *McCulloh et al., 2019*). In the shaded, moist forest, water is available nearly year-round, so we expected weaker coordination between leaves and roots (*Fortunel, Fine & Baraloto, 2012*). However, we expected stronger coordination among aerial traits related to light capture and use, which depend strongly on leaf biochemistry (*Wright et al., 2004*), leaf angle, branching pattern, and other aerial traits (*Valladares, Skillman & Pearcy, 2002*).

To test these hypotheses, we took advantage of the extensive knowledge of the ecology and functional strategies of seedlings in both forests and analyzed patterns of leaf, stem and root traits coordination among 94 tree-seedling species by means of multivariate correlation and network analysis.

## MATERIALS & METHODS

### Study sites

This study was conducted in two Mexican tropical forests: one located in the Montes Azules Biosphere preserve (16°04′N; 90°45′W) in Chiapas State, within the Lacandon region, and the other in the Chamela-Cuixmala preserve (19°30′N, 105°03′W) in Jalisco State, Mexico (Figs. S1, S2). Montes Azules Preserve covers an area of 331,200 ha, with an altitude range of 80 to 1,750 m a.s.l. This rainforest receives 3,000 mm of annual precipitation, distributed mostly between May and January, and a 3 month dry season with <100 mm of precipitation per month (*Ibarra-Manríquez & Martínez-Ramos, 2002*). The understory is highly shaded, receiving on average 1.83% of incident radiation, and soil fertility is highly spatially variable

(Table S1). The seedlings obtained from this tropical moist forest were distributed in an area of low hills over humic acrisol soils, with depths of 55 to 65 cm and moderate drainage (Table S1) (*Ibarra-Manríquez & Martínez-Ramos, 2002*). The Chamela-Cuixmala preserve is located on the Mexican Pacific coast and is dominated by a tropical dry forest that extends over low hills with an altitude range of 300 to 800 m above sea level. The mean annual precipitation ranges between 400 and 1,100 mm and occurs mainly from July to October, resulting in a 7-month dry season (November to June); most species lose their leaves in response to drought (*Lott, Bullock & Solis-Magallanes, 1987*). The forest understory is relatively open, receiving 13.7% of total incident radiation during the rainy season, while soils are generally stony and highly variable in depth (between 10 and 70 cm) and fertility (*Cotler, Durán & Siebe, 2002*). Given that soil nutrient content in our studies sites is highly variable spatially, and available nitrogen content overlapped between the two forest types, the main physical differences between the forests are related to water availability and light in the understory, as can be seen in Table S1.

## Plant material and traits

For the moist forest, during the rainy season, we collected seedlings of 43 dominant species (42 trees and one liana, see Table S2) that had no senescent cotyledons and at least one pair of leaves (average 21 seedlings per species, range 4–60). We carefully extracted the seedlings from the soil to preserve the integrity of all root tissues (following (*Paz, 2003*)). For the dry forest, we selected 52 woody dominant species (Table S2). It was not possible to extract the entire root system of seedlings because soils are thin and stony, so we collected seeds during the fruiting peak of each species. Seeds were stored in paper bags in the lab at room temperature until all of the seeds had been collected. We germinated the seeds in forest soil beds in a shade house, and five days after seedlings emerged, 16 seedlings per species were transplanted to 4.6 L (14 cm diameter × 30 cm tall) plastic bags with basal drainage, using forest soil as substrate. Seedlings were grown in a greenhouse for 3 months at 23.4 C (range 14–41 C) and 62% (40–83%) relative humidity, daily average photosynthetic photon flux of 805 millimoles/m$^2$/s and maintaining high volumetric soil water content (20%). These conditions fall within the range of variation detected in the mature forest floor during the rainy season (*Pineda-García, Paz & Meinzer, 2013*). See details of plant collection or propagation in Supplemental information.

The seedlings were washed carefully with tap water to remove soil particles from the roots. Maximum depth of the root system (MRD) was measured by extending the radicular system over a table and measuring the maximum length from the upper to the lower part with a ruler. Seedlings were divided into roots, stem, and leaves. Leaves were wrapped in moist paper towels and stored in sealed plastic bags for 12 h at 4 °C and after that time their saturated weight was obtained. Leaves where then digitized on a flatbed scanner (EPSON Expression 10000 XL, Japan). Roots were spread in a glass tray with water, and if necessary, cut to minimize overlap, and digitized with the same scanner. Leaf area was determined using Image J software (*Rasband, 2014*) and total root length and average root volume and diameter were measured with WinRhizo (*Arsenault et al., 2018*). A basal fragment of the stem of each seedling was cut under water and saturated in distilled water for 12 h.

**Table 1  Traits measured for each species in the tropical moist forest (Montes Azules), and the tropical dry forest (Chamela), Mexico.**

| Group | Trait | | Formula | Function | Units |
|---|---|---|---|---|---|
| Leaf axis | SLA | Specific leaf area | Leaf area/leaf dry weight | Growth rate, photosynthetic rate. Efficiency in resource acquisition. | $cm^2\ g^{-1}$ |
| | MPU | Minimum photosynthetic unit | Leaf area if entire or leaflet area if compound | Leaf cooling | $cm^2$ |
| | LTh | Leaf thickness | | Water storage capacity | mm |
| | LWC | Leaf water content | (Wet weight- dry weight)/dry weight | Water status | $g\ g^{-1}$ |
| Stem axis | WD | Wood density | Dry weight/wet volume | If low density, water storage, if high density, resistance against mechanical and herbivore damage | $g\ cm^{-3}$ |
| | SWC | Stem water content | (Wet weight- dry weight)/dry weight | Water storage capacity | $g\ g^{-1}$ |
| Root axis | SRL | Specific root length | Length of the roots/dry weight | Efficiency in resource acquisition. | $m\ g^{-1}$ |
| | MRD | Maximum root depth | | Deeper roots have access to a more stable soil water content | cm |
| | RTh | Root thickness | Average root diameter | Resources storage | cm |
| | RD | Root density | Dry weight/wet volume | Toughness, carbohydrates storage | $g\ cm^{-3}$ |

We obtained its saturated weight and its volume by the water displacement method. We removed the bark and obtained the wood weight and volume in the same way. Leaves, roots, wood and bark samples were then oven dried (leaves and roots for 48 h at 60 °C and wood and bark for 72 h at 70 °C) and we obtained their dry weight. Finally, we used these measurements to calculate the leaf, stem and root traits shown in Table 1. For the dry forest, we had leaf and stem data for 52 species and root data for 28 of those species, while in the moist forest all species had all traits (Table S2). We used the species mean values as data points for the statistical analyses.

## Statistical analysis

Before analysis, we assessed trait data normality independently for each forest. We performed log 10 transformations for all variables except for root density (RD) and maximum root depth (MRD) data from the dry forest (Chamela), and leaf thickness (LTh) and wood density (WD) data from the moist forest (Montes Azules). We assessed the influence of total plant biomass on each trait using linear regressions. In cases where this association was significant ($p < 0.05$), we used the residuals in lieu of the uncorrected trait value, to avoid the effect of size on traits. Hereafter we will refer to both original trait values and these residuals as "traits" for simplicity.

### Trait correlations

To assess the multivariate relationships among all traits, we performed principal component analysis (PCA) in R software v. 4.0.3 (*R Core Team, 2020*). To explore in detail the patterns of trait-trait associations within each forest, we performed Pearson correlations. Then, we

assessed whether correlations were significantly different between forests using the Fisher transformation of correlation to z score, using the *cocor* R package (*Diedenhofen & Musch 2015*).

Since trait relationships can be driven by phylogeny, we also performed a PCA using phylogenetically independent contrasts (PICs). For this, we computed PICs for each trait using the method described by *Felsenstein (1985)* using the *Ape* R package (*Paradis et al., 2017*). The phylogenetic tree of the species from each forest type was obtained according to *Qian & Jin (2016)*, setting branch length to 1 and resolving polytomies at random.

### Trait integration through network analysis

We used tools derived from network theory to assess trait coordination and to identify how traits were directly and indirectly related to each other. In this case, a network of trait relatedness represents functional traits as nodes, while significant correlations (negative or positive) are represented as links. We used the *igraph* R package (*Csardi & Nepusz, 2006*) to build the networks from the significant ($p < 0.05$) Pearson's correlations among traits. To measure the degree to which a network was divided into sub-groups (*i.e.*, modules) of highly connected traits (*i.e.*, modular network) we used the modularity and cluster spinglass functions from the *igraph* package (*Csardi & Nepusz, 2006*). These functions are based on simulated annealing to minimize the energy function of the network and calculate an optimal modularity value for each network (*Newman & Girvan, 2004*; *Reichardt & Bornholdt, 2006*). In general, the algorithm partitions all traits among distinct modules of highly connected traits and exports lists of trait membership within each module (*Yang, Algesheimer & Tessone, 2016*). This algorithm is widely used to measuring modularity in the context of community structure, mainly due to its robustness (*Newman & Girvan, 2004*). Modularity values range from $-1$ to 0.5 and are positive when the observed fraction of edges within the defined modules exceeds the fraction expected due to chance (*Messier et al., 2017*; *Flores-Moreno et al., 2019*). In networks with low modularity, traits interact weakly without any separate sub-groups, while higher modularity values indicate that there are traits that interact more strongly among themselves than with traits in other modules.

To measure the importance of the traits within each forest network, we considered three node descriptors: degree, betweenness centrality, and closeness centrality, using the *igraph* package (*Csardi & Nepusz, 2006*). Degree ($k$) is the number of traits with which a trait in the network is directly related. Because it is sensitive to network size, we divided degree values by the number of possible traits with which a trait in the network could be directly related ($n - 1$, where $n$ is the number of traits) to compare networks. For simplicity, we refer to this normlized measure only as degree. Betweenness centrality describes the role of a trait as a potential bridge between traits using the shortest distances connecting pairs of other traits. In contrast, closeness centrality quantifies the average length of the shortest path between a trait and all other traits in the network (*Antoniazzi Jr, Dáttilo & Rico-Gray, 2018*). Note that high values of all trait descriptors indicate greater importance within the networks. To better describe the importance of traits within each network, we used PCA based on the correlation matrix of metrics to summarize and combine the three-centrality metrics into a single value (*Dáttilo et al., 2016*; *Medeiros et al., 2018*). Finally, we performed

Person's correlations using PC1 values to test: (i) whether the most important traits in the dry forest were also the most important in the moist forest (*i.e.*, positive correlation) or, (ii) whether the most important traits in a forest type were the least important in the other forest (*i.e.*, negative correlation) or if the importance of traits were not related between dry and moist forests (*i.e.*, no correlation).

## RESULTS

### Bivariate and multivariate trait relationships in the moist and the dry forest

Multivariate analysis (Fig. 1, Table S3) showed a positive association between carbon investments in leaf, stem, and roots in both forests, showing a trade-off between organ density and water content. This is evidenced by a positive association between stem and leaf water contents (SWC, LWC) and a negative relationship of these traits with WD. Interestingly, although the signal of coordination with roots was present, it was less clear (Fig. 1). Specific root length (SRL) was directly related to leaf and stem water contents in the moist forest, while maximum root depth (MRD) and root density (RD) were related to WD in the dry forest. When using phylogenetic independent contrasts (PICs), most trait relationships remained (Fig. 1, Fig. S3). In the moist forest only, we found a positive correlation between SLA and SRL, traits which are associated with the efficiency in the acquisition of resources above ground and below ground, respectively ($r = 0.38$, $p = 0.01$, Fig. 2).

In addition, we found some correlations that were significantly different between forests (Fig. 2 and Fig. S4, Table S4). The strongest differences were found in the correlation between RTh and SRL ($z = 4.61$, $p = 0.000$), and between RTh and LWC ($z = 2.82$, $p = 0.005$), both of which were negative in the moist forest but positive in the dry forest. Other differences involved the MRD, which showed a significantly stronger positive relationship with WD ($z = 2.80$, $p = 0.005$) and a stronger negative relationship with SWC ($z = 2.19$, $p = 0.005$) in the dry forest (Fig. 2 and Table S4). There was also a slight difference between forests in the correlation of SLA with RTh (negative in the moist forest, positive in the dry forest, $z = 1.94$, $p = 0.053$), and SWC (positive in the moist forest and absent in the dry forest, $z = 1.9$, $p = 0.058$) (Fig. 2 and Table S4).

### Trait integration through network analysis

When considering forest networks formed by significant Pearson correlations (Table 2 and Fig. 3), the modularity values were intermediate in both forests (moist forest = 0.175; dry forest = 0.207) but were sufficiently high to identify two highly interrelated trait modules in each forest type (Fig. 3). In the moist forest, one module was formed by leaf and stem traits—LTh (leaf), MPU (leaf), LWC (leaf), and WD (stem)—while the other module was formed by leaf, stem, and root traits—SWC (stem), SLA (leaf), SRL (root), RD (root) and RTh (root). In the dry forest, one module was formed by leaf, stem, and root traits—LWC (leaf), MPU (leaf), LTh (leaf), MRD (root), SWC (stem), and WD (stem)—while the other module was formed only by root traits—RD (root), RTh (root), and SRL (root).
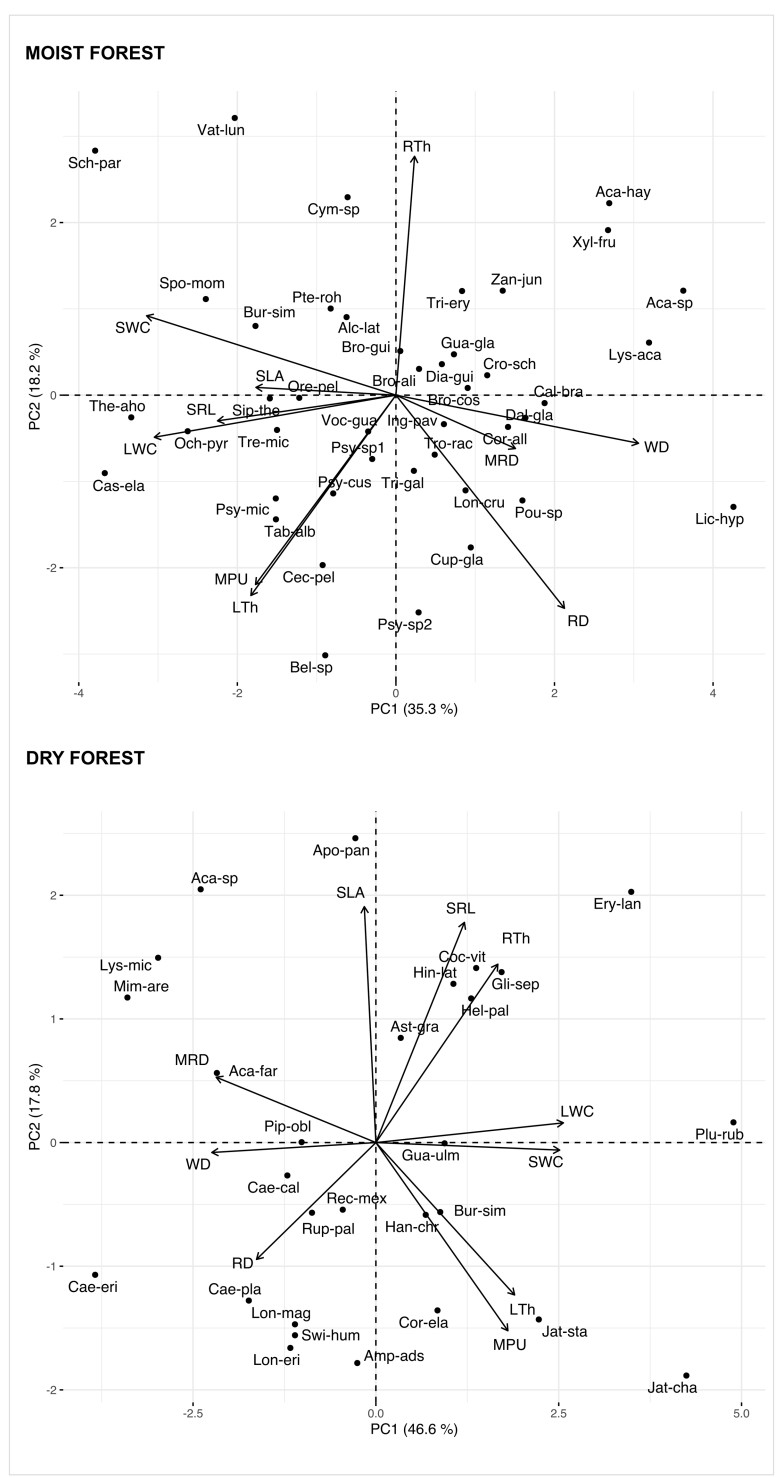

**Figure 1** **Principal Component Analysis (PCA) of seedlings traits for a moist tropical forest and a dry tropical forest in Mexico.** PCA is based on species' mean traits. SLA (specific leaf area); MPU (minimum photosynthetic unit); LTh (leaf thickness); LWC (leaf water content); WD (wood density); SWC (stem water content); SRL (specific root length); MRD (maximum root depth); RTh (root thickness); RD (root density). Acronyms for species names as in Table S2.

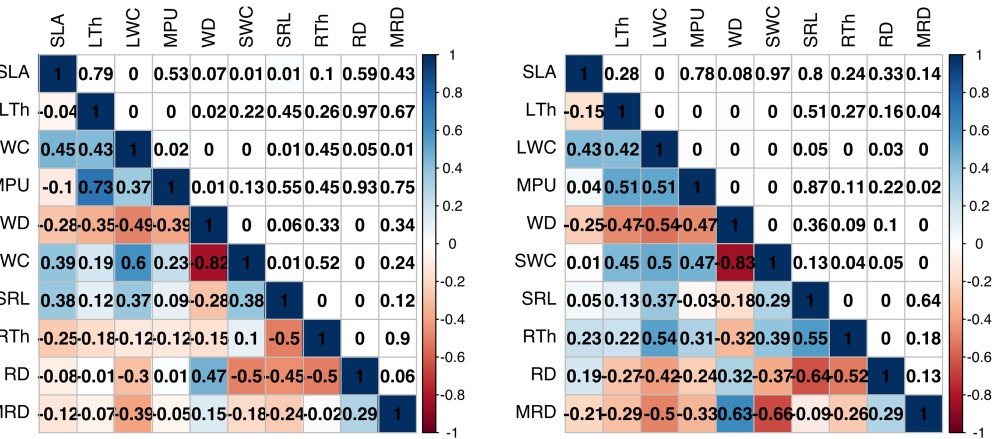

**MOIST FOREST**     **DRY FOREST**

**Figure 2**  Pearson's correlations between pairs of traits for seedlings of a tropical moist forest and a tropical dry forest. For each forest, correlation coefficients are shown in the lower triangle, and *P* values are shown in the upper triangle. Colors indicate the strength and sign of correlation (blue positive, red, negative). SLA (specific leaf area); MPU (minimum photosynthetic unit); LTh (leaf thickness); LWC (leaf water content); WD (wood density); SWC (stem water content); SRL (specific root length); MRD (maximum root depth); RTh (root thickness); RD (root density).

**Table 2**  Descriptors of trait's network centrality: degree, betweenness centrality, and closeness centrality for seedlings from moist and dry tropical forests.

|  | Moist forest | | | Dry forest | | |
|---|---|---|---|---|---|---|
|  | Betweenness | Closeness | Degree | Betweenness | Closeness | Degree |
| SLA | 0.00 | 0.60 | 0.33 | 0.00 | 0.50 | 0.11 |
| LTh | 0.00 | 0.56 | 0.33 | 0.00 | 0.64 | 0.55 |
| LWC | 0.41 | 0.82 | 0.77 | 0.48 | 0.90 | 0.88 |
| MPU | 0.00 | 0.56 | 0.33 | 0.00 | 0.64 | 0.55 |
| WD | 0.13 | 0.69 | 0.55 | 0.00 | 0.64 | 0.55 |
| SWC | 0.06 | 0.69 | 0.55 | 0.09 | 0.75 | 0.66 |
| SRL | 0.16 | 0.69 | 0.55 | 0.00 | 0.43 | 0.22 |
| RTh | 0.00 | 0.47 | 0.22 | 0.14 | 0.64 | 0.44 |
| RD | 0.08 | 0.60 | 0.44 | 0.06 | 0.60 | 0.33 |
| MRD | 0.00 | 0.47 | 0.11 | 0.00 | 0.64 | 0.55 |

**Notes.**

SLA, Specific leaf area; MPU, Minimum photosynthetic unit; LTh, Leaf thickness; LWC, Leaf water content; WD, Wood density; SWC, Stem water content; SRL, Specific root length; MRD, Maximum root depth; RTh, Root thickness; RD, Root density.

Principal component analysis (PCA) of the three centrality metrics (degree, betweenness centrality, and closeness centrality, Table 2) accounted for 93% of the variability in the dry forest and 85% in the moist forest. The highest PC1 scores in both forests corresponded to leaf water content (LWC, Fig. 4), indicating that this is a highly important trait, connected with many other traits by multiple direct and indirect pathways. Interestingly,

**MOIST FOREST**      **DRY FOREST**

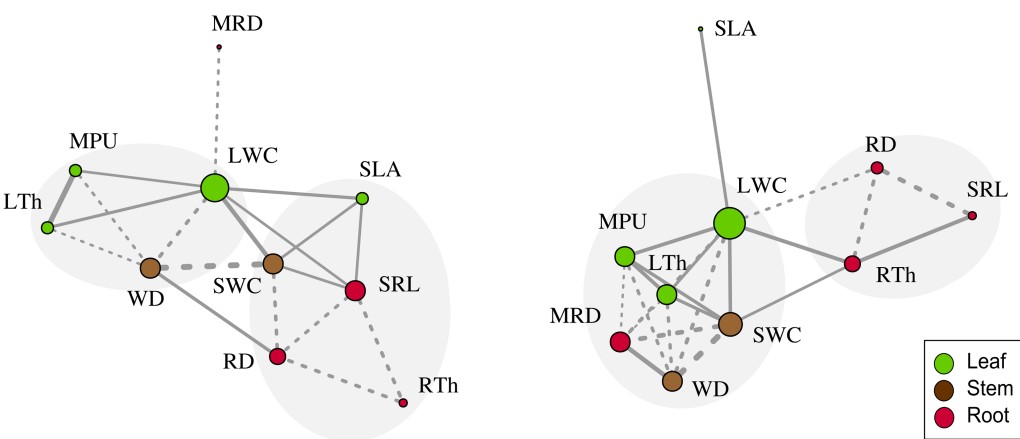

**Figure 3 Network trait correlations for a moist tropical forest and a dry tropical forest.** Nodes represent traits (size is proportional to the number of traits with which a trait in the network is related). Line width represents the strength of the correlation. Continuous and dashed lines represent positive and negative correlations, respectively. Gray shades identify trait modules. SLA (specific leaf area); MPU (minimum photosynthetic unit); LTh (leaf thickness); LWC (leaf water content); WD (wood density); SWC (stem water content); SRL (specific root length); MRD (maximum root depth); RTh (root thickness); RD (root density).

the correlation between the first principal component (PC1) scores of each forest centrality PCA showed that the most important traits in one forest type were not necessarily the most important in the other forest ($r^2 = 0.55$; $p = 0.09$), indicating that the most important linking traits differed between the two environments (Fig. 4). For example, while MRD showed marginal importance in the moist forest (only weakly coordinated with LWC), this same trait was highly important in the dry forest and showed multiple relationships with leaf and stem traits (Figs. 3 and 4). Likewise, while specific root length (SRL) had marginal importance in the dry forest, this trait had high importance in the moist forest (Fig. 4).

## DISCUSSION

### Coordination is similar between forests, but trait relationships are not

We hypothesized that water shortage in the dry forest selects for stronger coordination among leaf, stem, and root traits, while shade in the understory of the wet forests favors strong coordination only between leaf and stem traits. However, we found no support for such hypothesis; in both forests we detected modules that included traits from all three organs. This is similar to findings by *Flores-Moreno et al. (2019)*, who also found high connectivity across trait networks within and between tissue types in both tropical and temperate forests. A novel contribution from our study is the use of network analysis to study inter-trait coordination not only between stem and leaf traits, but also with root traits. Interestingly, the finding that the pattern of coordination of key traits was different

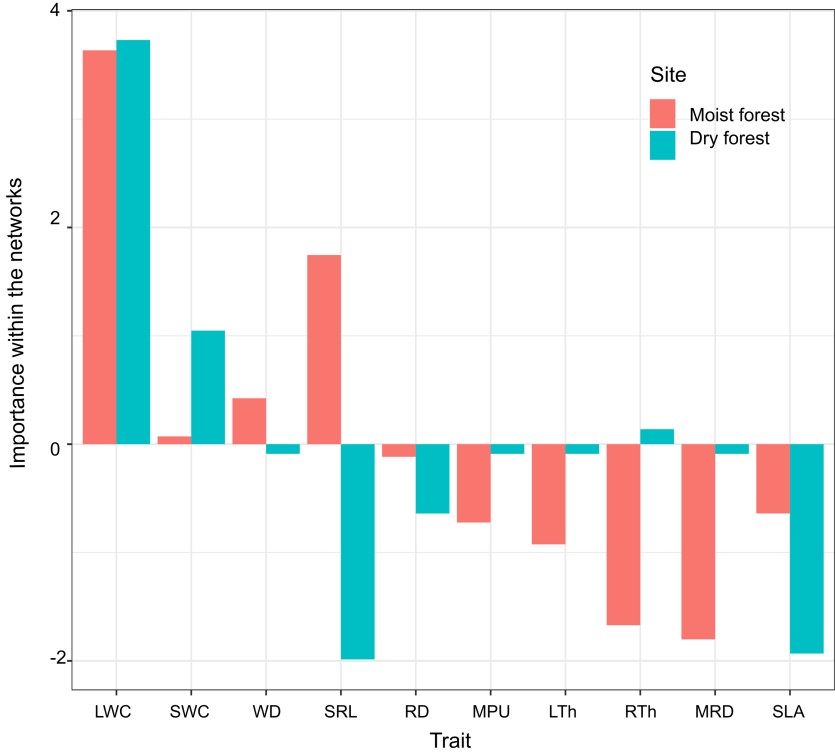

**Figure 4  Importance of functional traits within the network of trait correlations in the moist and dry tropical forests studied.** Trait importance was measured by summarizing and combining three-centrality metrics (degree, closeness, and betweenness) into a single value (first principal component score - PC1, which accounted for 93% of the variability in the dry forest and 85% in the moist forest). SLA (specific leaf area); MPU (minimum photosynthetic unit); LTh (leaf thickness); LWC (leaf water content); WD (wood density); SWC (stem water content); SRL (specific root length); MRD (maximum root depth); RTh (root thickness); RD (root density). Moist forest values (red), dry forest values (blue).

between the dry and the moist forest supports the hypothesis that the selective value of the same traits differs under different conditions (*Dwyer & Laughlin, 2017*; *Flores-Moreno et al., 2019*). For example, in the moist forest we found weak coordination between the morphological efficiency to capture resources above ground and below ground (*i.e.*, direct correlation between SLA-SRL, Fig. 2), but these traits were not correlated in the dry forest. The existence of a growth-defense trade-off has been well described among moist forest species (*Poorter & Bongers, 2006*); fast growth in gaps is promoted by cheap, short-lived, and physiologically active leaves (indicated by high SLA), while high survival in the forest understory is enhanced by the formation of long-lived, tough leaves that reduce herbivory, mechanical damage, or leaf turnover (*Poorter & Bongers, 2006*). The coordination between SRL and SLA detected in our study (Figs. 2 and 3), suggests that the growth-defense trade-off involves both above- and below-ground morphological traits. Species with high SLA that are adapted to rapidly acquire light, seem to also have roots that are capable of absorbing soil resources efficiently. Meanwhile, species with high investment in structural defense of leaves (low SLA), tend to also have structurally well-defended roots (low SRL).
However, the weak SLA-SRL correlation ($r = 0.38$, $p = 0.01$) and the inconsistency of the relationship between these traits in published literature (*Weigelt et al., 2021*) may be related to the fact that SRL depends on root density (RD) and root thickness (RTh), and plants can construct roots with many combinations of these values. In addition, RTh is also related to symbiosis with mycorrhizal fungi. Consequently, roots vary not only along a conservation-acquisition trade-off, but also, and orthogonally, along a collaboration-do it yourself trade-off (*Bergmann et al., 2020*). The role played by symbiosis in root traits (SRL and RTh) needs further investigation in this moist forest.

The lack of SLA-SRL coordination in the dry forest may also reflect the fact that the development of dense tissues with high carbon investment is not necessarily the predominant strategy to deal with drought. Commonly, drought-deciduous species have short-lived, low-cost leaves with high SLA, coupled with low-density stems and thick root tissues containing high water and carbohydrate reserves (*Powers & Tiffin, 2010*; *Pineda-García, Paz & Tinoco-Ojanguren, 2011*; *Palomo-Kumul et al., 2021*). Conversely, maximum root depth (MRD) was coordinated with leaf and stem traits in the dry forest, but not in the moist forest (Fig. 3). The relevance of MRD in this and other dry forests is not surprising, given the importance of this trait for seasonal water uptake in seedlings in arid environments (*Ackerly, 2004*; *Maeght, Rewald & Pierret, 2013*), especially during drought periods (*León et al., 2011*; *Padilla & Pugnaire, 2007*). Furthermore, the coordination of dense stems with deep roots in the dry forest (Figs. 2 and 3) highlights the importance of a more permanent source of water when stem storage capacity is low (*Schwinning & Ehleringer, 2001*; *Hasselquist, Allen & Santiago, 2010*; *Paz, Pineda-García & Pinzón-Pérez, 2015*). Likewise, thick roots (RTh) were directly related with high SRL and leaf water content (LWC) in the dry forest (and inversely in the moist forest), an indication that RTh may be related to water economy in the dry forest. Commonly in dry forests, roots store important amounts of water in their tissues (*Paz, Pineda-García & Pinzón-Pérez, 2015*), while in moist forests thick roots are commonly fibrous.

Although our moist forest seedlings were obtained directly from the field while dry forest seedlings were grown in pots, we are confident that this did not affect root trait measurements, particularly MRD. First, in our study of dry forest plants, we rarely observed roots reaching the bottom or lateral edges of the pots, indicating the soil volume did not impede root growth in a specific direction. Second, in a previous study where MRD was measured in four neotropical forests with the same methodology, extracting seedlings from the ground, roots were deeper in the site with the longest dry season (*Paz, 2003*), similar to our study.

## Trait centrality

When considering the traits' centrality values (the relative importance of each trait as a connecting node), we found that LWC was the most important trait in both forests (Fig. 4), reflecting the key role of leaf hydration in plant growth and many other physiological processes, regardless of forest type. Previous studies have proposed that in addition to leaf turgor, maintenance of leaf water content may be critical for growth processes such as cell elongation and division, as well as for leaf gas exchange (*Bartlett, Scoffoni & Sack, 2012*). In
addition, leaf water content has been found to be associated with hydraulic traits at the stem level, such as stem water content and stem conductivity (*Pineda-García et al., 2015*) and is a good predictor of growth and survival under dry conditions in other tropical forests (*Cifuentes et al., 2020*). The frequent correlations of wood density with hydraulic and leaf traits in adult trees (*Santiago et al., 2004*; *McCulloh et al., 2011*; *Greenwood et al., 2017*, among others) have led to the idea that wood density could represent a central trait affecting the stem and leaf economy (*Chave et al., 2009*; *McCulloh et al., 2011*; *Méndez-Alonzo et al., 2013*). However, strikingly, in our study we detected strong evidence suggesting that among tropical seedlings, LWC may be another key functional trait; this is an idea worth testing in other tropical forests.

Interestingly, SWC was the second most important trait in the dry forest, but not in the moist forest. This may be explained by the close association of SWC with the water storage capacity *versus* soil vertical foraging and water exploitation *versus* drought tolerance trade-offs described in tropical seasonally dry forest (*Paz, Pineda-García & Pinzón-Pérez, 2015*; *Pineda-García et al., 2015*). Drought avoiders, which have high photosynthetic rates, xylem hydraulic conductivity, and growth rate when water is available (*Pineda-García et al., 2015*), maintain a narrow safety margin between plant water potential and $P_{50}$ (the potential that would induce 50% of hydraulic conductivity loss by the formation of emboli) (*Pineda-García, Paz & Meinzer, 2013*; *Markesteijn et al., 2011b*). Given that these species tend to have shallow roots, they have evolved a great capacity to store water in the stem and leaves, which allows them to survive as the soil desiccates during the dry season (*Paz, Pineda-García & Pinzón-Pérez, 2015*). On the contrary, drought-tolerant species, which have lower photosynthetic rates, xylem hydraulic conductivity, and growth rate, have a denser stem with limited water storage capacity (low SWC) (*Pineda-García et al., 2015*; *McCulloh et al., 2019*). Due to their lower capacity to decouple hydraulically from the soil, these dense-tissue species are associated with deep roots that penetrate deeper into the soil and rock interstices and rely on a more constant, although unsaturated, soil water content (*Schwinning & Ehleringer, 2001*; *Padilla & Pugnaire, 2007*; *Zhou et al., 2020*). Thus, differential responses to drought explain the importance of SWC, a trait strongly involved in water economy in the dry forest.

As expected, SLA had very low centrality in the dry forest (Fig. 4), where competition for light is likely not as strong as competition for water, and where SLA is strongly related to non-morphological traits such as leaf phenology. However, the low centrality of SLA in the moist forest is intriguing. This lack of finely tuned connection between a central trait in the leaf economy spectrum and other leaf and stem traits suggests that different combinations of leaf traits and plant architectures can yield similar capacities for growth in shaded forest (*Valladares, Skillman & Pearcy, 2002*). This hypothesis needs further investigation in our study system. Contrary to expectation, SRL, a trait typically claimed to be a key morphological determinant of the efficiency of carbon investment in water absorption, was poorly connected in the dry forest, but highly connected in the wet forest (Fig. 4). Although it is possible that this was due to differences in soil nutrients between forests, this seems unlikely since nutrient levels overlap strongly between the study sites. Our field observations suggest that in the dry forest, high values of SRL may be indicative of different

functions depending on the species: fine roots deploying large absorptive surfaces, or thick roots with high water and low carbon contents acting as water storage more than for water absorption. Together, the lack of SRL centrality and the clear correlations of MRD with leaf and stem traits in the dry forest suggest that in habitats with a high risk of drought at the seedling stage, developing deep roots has a higher selective value than developing thin, efficiently absorptive roots (*Padilla & Pugnaire, 2007*; *León et al., 2011*; *Paz, Pineda-García & Pinzón-Pérez, 2015*). Conversely, in the moist forest, in the context of strong competition for light, SRL and WD are important under a growth-survival trade-off, where species that acquire soil resources efficiently grow fast (high SRL) and have low-density stems (low WD), and vice versa (*Poorter, 2009*; *Pineda-García et al., 2015*).

## CONCLUSIONS

We found that in the two forests we studied, which differ in precipitation and seasonality, the level of coordination among leaves, shoots, and roots in seedlings was similar, but the most functionally connected traits were different. In the dry forest, the most central traits were all related to water storage (LWC, SWC, RTh), while in the moist forest they were related to the capacity to store water in leaves (LWC), root efficiency to capture resources (SRL), and stem toughness (WD). Our findings suggest that, along with precipitation, there is a shift in the relative importance of mechanisms to face the most limiting resource. In the dry forest, this is the water storage capacity, soil vertical foraging, and water exploitation-drought tolerance trade-offs. In the moist forest, the growth-survival trade-off is most important. However, further studies of leaf, stem, and root coordination including different ontogenetic stages and multiple sites over environmental gradients are needed to clarify whether plants respond to limiting resources under a "whole plant" strategy, or whether limiting resources or phylogenetic constraints act on different plant organs independently.

## ACKNOWLEDGEMENTS

L.S.V. and H.P. appreciate the support from personnel of the Chamela Biological Station, UNAM, and Chajul Village, Mexico. The authors thank Lynna Kiere for English language editing.

### Funding

This work was supported by Consejo Nacional de Ciencia y Tecnología, México (Grant CB240607 to Horacio Paz), and Dirección General de Asuntos de Personal Académico (DGAPA) from Universidad Nacional Autónoma de México (UNAM), (Grant no. IN207618 to Horacio Paz). Lucía Sanaphre-Villanueva received a postdoctoral scholarship from DGAPA-UNAM and CONACYT. Horacio Paz received sabbatical support from DGAPA, UNAM. The funders had no role in study design, data collection and analysis, decision to publish, or preparation of the manuscript.

## Grant Disclosures

The following grant information was disclosed by the authors:

Consejo Nacional de Ciencia y Tecnología, México: CB240607.

Universidad Nacional Autónoma de México (UNAM): IN207618.

DGAPA-UNAM and CONACYT.

DGAPA-UNAM and CONACYT.

DGAPA.

UNAM.

## Competing Interests

The authors declare there are no competing interests.

## Author Contributions

- Lucía Sanaphre-Villanueva conceived and designed the experiments, performed the experiments, analyzed the data, prepared figures and/or tables, authored or reviewed drafts of the article, and approved the final draft.
- Fernando Pineda-García performed the experiments, authored or reviewed drafts of the article, and approved the final draft.
- Wesley Dáttilo analyzed the data, prepared figures and/or tables, authored or reviewed drafts of the article, and approved the final draft.
- Luisa Fernanda Pinzón-Pérez performed the experiments, authored or reviewed drafts of the article, and approved the final draft.
- Arlett Ricaño-Rocha performed the experiments, authored or reviewed drafts of the article, and approved the final draft.
- Horacio Paz conceived and designed the experiments, performed the experiments, authored or reviewed drafts of the article, and approved the final draft.

## Data Availability

The mean functional plant traits for 94 tree species from a tropical dry and a moist forest are available in the Supplementary File and at Zenodo: Paz, H. (2022). Data for: ''Above and below ground trait coordination in tree seedlings depend on the most limiting resource: a test comparing a wet and a dry tropical forest in Mexico'' by L. Sanaphre-Villanueva, F. Pineda-Garcia, W. Dattilo, L. F. Pinzon-Perez, A. Ricaño Rocha, H. Paz. [Data set]. Zenodo. https://doi.org/10.5281/zenodo.6512978.

## Supplemental Information

Supplemental information for this article can be found online at http://dx.doi.org/10.7717/peerj.13458#supplemental-information.

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
