# Peer review of "Above- and below-ground trait coordination in tree seedlings depend on the most limiting resource: a test comparing a wet and a dry tropical forest in Mexico"

_PeerJ, doi:10.7717/peerj.13458_

## Round 0.1 · original submission · Major Revisions

Please revise your manuscript to address the concerns of the reviewers.

Reviewer 1 ·

Basic reporting

no comment

Experimental design

Data analysis should be improved.

Validity of the findings

In this manuscript, the authors compared the aboveground and belowground traits of a dry forest and a wet forest, and want to find if the traits differences are formed by the environmental stress. The trait differences between two forests are listed clearly in the manuscript, but they are not correlated with the environmental factors analysis. The authors cannot make conclusion about if the traits differences are necessarily formed by the environmental factors or which factor simply. The analysis also did not show which environmental factor is the most limiting resource influenced the species traits. My suggestion is that the correlation analysis between environmental factors and traits are needed instead of simple traits differences comparison. To improve the manuscript, the theoretical frame about how environmental factors limiting traits variation should be also strengthen.

Additional comments

The manuscript could be improved by strengthen the theoretical frame and data analysis.

Reviewer 2 ·

Basic reporting

In this paper the authors analyze the importance of functional traits in seedlings of 94 tropical tree species in two contracting ecosystems. It is an interesting and current topic for understanding plant responses to changes in the environment. The introduction documents the topic adequately; however, throughout the paper the citations used are not very recent and more than five years old, there are only 3 citations from 2020-2021. So I recommend updating the introduction and discussion, some articles that could support are the following.


Stem functional traits, not just morphology, explain differentiation along the liana–tree continuum
LS Santiago - Tree Physiology, 2021
El Niño-Southern Oscillation affects the water relations of tree species in the Yucatan Peninsula, Mexico J Palomo-Kumul, M Valdez-Hernández, GA Islebe… - Scientific Reports, 2021
Hydraulic traits of Neotropical canopy liana and tree species across a broad range of wood density: implications for predicting drought mortality with models ME De Guzman, A Acosta-Rangel, K Winter… - Tree Physiology, 2021
Unraveling the relative role of light and water competition between lianas and trees in tropical forests: A vegetation model análisis F Meunier, H Verbeeck, B Cowdery, SA Schnitzer… - Journal of Ecology, 2021

Experimental design

no comment

Validity of the findings

no comment

Additional comments

I recommend replacing table 2 with table S1, since I consider it very important to have the list of species studied in the paper. Table 2 should be placed in the supplementary material.
I consider it necessary to add in Table S1, which traits were measured for each species? Since, in the methodology they mention that only root data were obtained for 28 species in the dry forest.

Reviewer 3 ·

Basic reporting

Figures must be supplied at a greater resolution.

Experimental design

The description of how traits were measured is completely missing. I understand that this detailed description is in a supplemental material, but a brief description should be included in the main text. Furthermore, statistical analyses can be summarized in the main text, and explain in more detail in the supplemental materials.

Validity of the findings

This work seeks to understand trait coordination and trade-offs among plant organs in species from two tropical forests: one dry and one moist. The authors studied a large number of species and traits related to plant performance and found coordination in both sites, but the traits involved in those coordinations differed between sites. This is an important contribution to understanding plant function in tropical forests, demonstrating that trait coordination in the dry site involved traits related to water acquisition and storage, as water is the main limiting resource in those forests.

Data have not been provided

Additional comments

Major comments
The introduction has enough background information to understand the motivation behind this work. I suggest in the PDF that the authors check other papers on the coordination of traits among organs. Also, given that coordination and trade-offs are frequently used throughout the manuscript I recommend that the authors define these terms at first mention in the introduction.
The description of how traits were measured is completely missing. I understand that this detailed description is in a supplemental material, but a brief description should be included in the main text. Furthermore, statistical analyses can be summarized in the main text, and explain in more detail in the supplemental materials.
Results are concise and well describe. I added a comment in the discussion related to the relationship between SLA and SRL. Please revise that the growth-defense trade-off is appropriate for explaining these results.
Minor comments
I have included minor comments directly in the PDF attached.
Please supply images of better resolution.
Please include data availability statement and mention if the data is published elsewhere or if it will be published as supplemental materials.
I also recommend that the authors include pictures of the plants/sites in the supplemental materials.

Annotated reviews are not available for download in order to protect the identity of reviewers who chose to remain anonymous.

---

## Round 0.2 · accepted · Accept

Thank you for your careful revisions in light of the reviewers' comments.